

# Retrieval of Black Carbon Aerosol Surface Concentration Using Integrated MODIS and AERONET Data

Xingxing Jiang[1], Yong Xue[2], Mariarosaria Calvello[3], Shuhui Wu[1], Pei Li[1]

[1]School of Environment and Spatial Informatics, China University of Mining and Technology, Xuzhou 221116, China
[2]School of Emergency Management, Nanjing University of Information Science and Technology, Nanjing 210044, China
[3]Institute of Methodologies for Environmental Analysis, National Research Council, Tito Scalo 85050, Italy

*Correspondence to*: Yong Xue (yxue@nuist.edu.cn)

**Abstract.** Black Carbon (BC) is a carbonaceous aerosol that strongly absorbs solar radiation. The high emissions of these highly absorbent particles exacerbate regional air quality and pose significant threats to global climate, both in the short and
long term. Therefore, accurately quantifying the spatial distribution of BC is crucial for improving regional air quality and mitigating the climate change impacts driven by human activities. In this study, we developed a novel algorithm for retrieving BC surface concentration using MODIS and AERONET data. The algorithm first determined the seasonal background aerosol model using the K-means clustering method, based on AERONET V3 daily products. It then employed the Maxwell–Garnett effective medium approximation model to calculate the complex refractive index of the internally
mixed aerosols and used the 6SV2.1 radiative transfer code to establish lookup tables for optimal BC fraction and column concentration estimation. Subsequently, the column concentration data were converted to surface concentration using a conversion coefficient derived from MERRA-2. Finally, the retrieved MODIS BC surface concentration was validated with in-situ Aethalometer measurements. The validation showed a correlation coefficient (R) of 0.727, a root mean square error (RMSE) of 0.353, a mean absolute error (MAE) of 0.211, and a linear fit function of y = 0.718x + 0.015. These statistical
parameters outperform those obtained from MERRA-2 BC data, demonstrating the superior performance of the proposed algorithm in this study area.

**Key words:** Black carbon, Surface concentration, retrieval, MODIS

## 1 Introduction

Black carbon aerosol (BC) particles have important impacts on the global climate, air pollution, and human health
(Ramanathan and Carmichael, 2008). According to the latest report of the Intergovernmental Panel on Climate Change (IPCC) (Everett et al., 2022), the overall climate effect of aerosols is cooling, but the BC particles therein have a significant warming effect, and play an extremely important role in promoting glacier melting in the Arctic region (Flanner, 2013). In addition, BC emitted from human activities also significantly affects air quality (Cao et al., 2007). It is one of the main components of fine particulate matters (PM2.5) (Cai et al., 2020), which has a great impact on human health and is
considered an important factor leading to obesity (Guo et al., 2022). Therefore, it is of great research significance to obtain



accurate and reliable BC spatiotemporal distribution, especially BC surface concentration data that is extremely relevant to human activities.

Currently, the most common method to obtain the chemical composition of BC relying on in-situ measurements, such as Aethalometers (AE31/33) (Pavese et al., 2020), aerosol mass spectrometer (Wang et al., 2018), scanning electron microscopy (Brodowski et al., 2005)., etc. These methods are accurate and reliable but can only obtain BC concentrations in specific areas. In addition, using chemical transport models to simulate BC concentrations is also an important method (Xu et al., 2020). Still, the current assimilation products have low spatial resolution and the results are highly uncertain due to the deviation of model input data from actual atmospheric conditions (Sato et al., 2003).

Over the past 20 years, obtaining aerosol physical and chemical properties based on ground-based remote sensing and satellite remote sensing methods has been a hot topic in atmospheric science research (Remer et al., 2024). In ground-based remote sensing, some scholars used the Aerosol Robotic Observation Network (AERONET) to invert the BC-dominated absorption component based on the observation results of the complex refractive index (RI), combined with the three aerosol characteristics of BC, ammonium sulfate, and aerosol water (Sato et al., 2003; Schuster et al., 2005). Later, the single scattering albedo (SSA) was introduced based on the RI, and a five-component inversion model was established (Wang et al., 2013). On this basis, Xie et al. (2017) introduced the log-normal volume size distribution, and then using fine mode volume concentration and coarse mode volume concentration, combined with aerosol optical depth (AOD) correction, successfully separated the proportion of coarse and fine particles, and simulated the concentration and volume distribution of seven different aerosol types. Zhang et al. (2024) simultaneously obtained the long-term series BC column concentration of global AERONET stations based on the Generalized Retrieval of Aerosol and Surface Properties (GRASP)/Component algorithm (Dubovik et al., 2011). However, ground-based remote sensing cannot accurately describe the spatial variation of BC. Therefore, multi-temporal and wide-area observations using satellite remote sensing have the potential to monitor the large-scale spatiotemporal variation of BC. Some studies have preliminarily proposed BC concentration inversion algorithms based on satellite data, such as applying the GRASP algorithm to polarized satellite data such as Polarization and Directionality of the Earth's Reflectances (POLDER) (Bao et al., 2019; Li et al., 2019, 2020) and Directional Polarimetric Camera (DPC) instruments (Li et al., 2022). Based on Moderate Resolution Imaging Spectroradiometer (MODIS) data, the BC column concentration was estimated in China using the lookup table method (LUT) (Bao et al., 2020). Based on the geostationary satellite Himawari8 data, the hourly BC surface concentration in North China was estimated using the critical reflectance method (Bao et al., 2023). In this paper, we utilized MODIS data, long-term AERONET aerosol optical property observation data, considering the seasonal differences of background aerosols (BAs) across the study area. The K-means method was applied to categorize the optical properties of BAs for different seasons, and multiple LUTs were established with varying BC fraction. This enables the development of a novel BC surface concentration inversion algorithm tailored to the region. Given that MODIS has provided a substantial volume of long-term data, this new algorithm holds significant potential for investigating long-term spatiotemporal changes in BC concentrations. The Section 2 describes data source; Section 3 illustrates the methodology, including forward model, calculation strategy for physical properties of internal mixed



aerosols, inverse method, and sensitivity studies; Section 4 contains the retrieved MODIS BC surface concentration results, validation, and uncertainty analysis. Section 5 represents conclusion.

## 2 Data

### 2.1 MODIS data

MODIS has been recording data on the Aqua and Terra satellites launched by NASA and has been providing a large amount
of observations since 1999 (Remer et al., 2005). This study used MODIS data from November 2023 to June 2024 in the study area of 5°E - 20°E, 30°N - 50°N, including three types of datasets: MO/YD02 (L1B data), MO/YD03 (Geolocation data), and MO/YD04 (AOD data based on Dark Target algorithm (DT), 0.55µm). The DT used the linear relationship between the surface reflectance of 0.47µm, 0.66µm and 2.12µm to retrieve AOD. This product has been widely used in atmospheric remote sensing and climate change research due to its reliable accuracy and long time series. The DT used the
linear relationship between the surface reflectance of 0.47µm, 0.66µm and 2.12µm to retrieve AOD (Levy et al., 2013). This product has been widely used in atmospheric remote sensing and climate change research due to its reliable accuracy and long time series. These datasets can be obtained from this website (https://ladsweb.modaps.eosdis.nasa.gov/).

### 2.2 AERONET data

AERONET is the world's most widely used ground-based aerosol physical characteristics observation network, providing
long-term aerosol optical and physical property observation data from thousands of stations for nearly 30 years (Dubovik et al., 2000). This study used the AERONET V3 daily dataset of 32 stations in the study area to obtain BAs characteristic data, which was used as aerosol model input data in the atmospheric radiation transfer model. The locations of these sites are shown as red dots in Fig. 1, and the detailed site information is shown in Table 1. This dataset can be downloaded at (https://aeronet.gsfc.nasa.gov/).

### 2.3 AE33 data

AE33 aethalometer is based on the principle of light absorption and quantifies BC surface concentration by measuring the light absorption characteristics of aerosol samples at multiple wavelengths (Yus-Díez et al., 2021). The instrument typically conducts real-time continuous light absorption measurements at seven wavelengths, ranging from ultraviolet to near-infrared, allowing it to distinguish between different sources of BC and aerosol components, thus improving data accuracy (Rajesh
and Ramachandran, 2018). In this study, we used the BC surface concentration data from 6 sites equipped with AE33 and located in the study area, with the measurement wavelength at 637nm. The locations of the 6 AE33 sites are shown in Fig. 1, and the detailed site information is shown in Table 1. The AE33 BC surface concentration data can be obtained from this website (https://ebas.nilu.no/data-access/).




## 2.4 MERRA-2 data

MERRA-2 is a global atmospheric reanalysis dataset developed by NASA. It is specifically designed to provide high-quality historical datasets for the study of atmospheric and climate processes (Gelaro et al., 2017). In this paper, we used water vapor and ozone data to correct the absorption of MODIS L1B data, and BC column concentration and surface concentration data were used to obtain a priori ratio and for comparison. The MERRA-2 datasets can be downloaded from (https://search.earthdata.nasa.gov/).

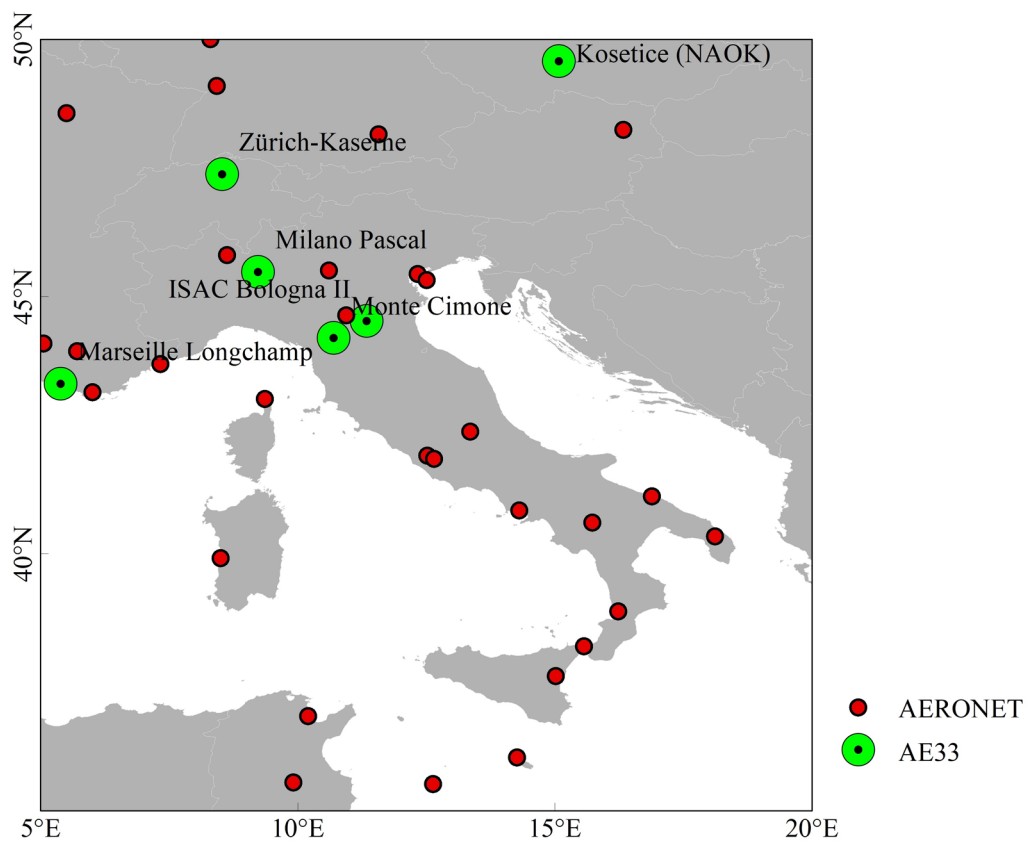


**Fig. 1.** Location distribution of AERONET and AE33 ground observation stations. The small red dots, large green dots, and text in the figure represent AERONET sites, AE33 sites, and AE33 site names, respectively.

**Table 1.** AE33 and AERONET ground observation stations parameters.

| Station type | Station name | Longitude (°) | Latitude (°) | Altitude (m) |
|---|---|---|---|---|
| AE33 | Monte Cimone | 10.70 | 44.19 | 2165 |
| | ISAC Bologna II | 11.34 | 44.52 | 54 |
| | Milano Pascal | 9.23 | 45.48 | 118 |
| | Zürich-Kaserne | 8.53 | 47.38 | 409 |





| | | | | |
|---|---|---|---|---|
| | Marseille Longchamp | 5.39 | 43.31 | 73 |
| | Kosetice (NAOK) | 15.08 | 49.57 | 538 |
| AERONET | AAOT | 12.51 | 45.31 | 10 |
| | Bari_University | 16.89 | 41.11 | 12 |
| | Ben_Salem | 9.91 | 35.55 | 130 |
| | Bure_OPE | 5.51 | 48.56 | 393 |
| | Carpentras | 5.06 | 44.08 | 107 |
| | Ersa | 9.36 | 43.00 | 80 |
| | ETNA | 15.02 | 37.61 | 736 |
| | Gozo | 14.26 | 36.03 | 111 |
| | IMAA_Potenza | 15.72 | 40.60 | 770 |
| | IMC_Oristano | 8.50 | 39.91 | 10 |
| | ISDGM_CNR | 12.33 | 45.44 | 20 |
| | Ispra | 8.63 | 45.80 | 235 |
| | Karlsruhe | 8.43 | 49.09 | 140 |
| | Lamezia_Terme | 16.23 | 38.88 | 8 |
| | Lampedusa | 12.63 | 35.52 | 45 |
| | LAQUILA_Coppito | 13.35 | 42.37 | 656 |
| | Lecce_University | 18.11 | 40.34 | 30 |
| | Mainz | 8.30 | 50.00 | 150 |
| | Messina | 15.57 | 38.20 | 15 |
| | Modena | 10.95 | 44.63 | 56 |
| | Munich_University | 11.57 | 48.15 | 533 |
| | Napoli_CeSMA | 14.31 | 40.84 | 50 |
| | OHP_OBSERVATOIRE | 5.71 | 43.94 | 680 |
| | Rome_La_Sapienza | 12.52 | 41.90 | 75 |
| | Rome_Tor_Vergata | 12.65 | 41.84 | 130 |
| | Sirmione_Museo_GC | 10.61 | 45.50 | 86 |
| | Toulon | 6.01 | 43.14 | 50 |
| | Tunis_Carthage | 10.20 | 36.84 | 10 |
| | Venise | 12.51 | 45.31 | 10 |
| | Vienna_BOKU | 16.33 | 48.24 | 266 |
| | Villefranche | 7.33 | 43.68 | 130 |



## 3 Methodology

### 3.1 Forward model

In this study, we used the 6SV2.1 model (Vermote et al., 2016), which is widely used in current aerosol remote sensing inversion. When the zenith angle does not exceed 75°, the estimation error of top of atmosphere reflectance (TOA) does not exceed 0.4% (Jiang et al., 2022). The 6SV2.1 model equation is as follows:

$$\rho_{TOA}\left(\theta_s,\theta_v,\varphi,\tau,f_{BC},BAs\right)=$$
$$T_g\left[\begin{array}{l}\rho_{atm}\left(\theta_s,\theta_v,\varphi,\tau,f_{BC},BAs\right)+\\ T_1\left(\theta_s,\tau,f_{BC},BAs\right)T_2\left(\theta_v,\tau,f_{BC},BAs\right)\rho_s\Big/\left(1-\rho_s S\left(\tau,f_{BC},BAs\right)\right)\end{array}\right] \tag{1}$$

In the Eq. (1), $\theta_s$, $\theta_v$, $\varphi$, $\tau$ and $f_{BC}$ denote solar zenith angle, satellite zenith angle, relative azimuth angle, AOD, and BC volume fraction, respectively. $\rho_{TOA}$, $\rho_{atm}$, $\rho_s$, $S$, $T_g$, $T_1$, and $T_2$ represent TOA, atmospheric path reflectance, surface reflectance, atmospheric spherical albedo, gaseous transmission, downward atmospheric transmission, and upward atmospheric transmission, respectively.

### 3.2 Estimation of optical properties of mixture aerosol

Maxwell−Garnett effective medium approximation model (MG-MEA) is used to estimate the Maxwell-Garnett dielectric function of aerosol mixtures (Schuster et al., 2005). In this paper, the schematic diagram of the mixture including BC surrounded by BAs is shown in Fig. 2. The MG-EMA equation is as follows:

$$\varepsilon_{MG}=\left[1+3f_{BC}\left(\left(\varepsilon_{BC}-\varepsilon_{BAs}\right)/\left(\varepsilon_{BC}+2\varepsilon_{BAs}\right)\right)/\left(1-f_{BC}\left(\varepsilon_{BC}-\varepsilon_{BAs}\right)/\left(\varepsilon_{BC}+2\varepsilon_{BAs}\right)\right)\right] \tag{2}$$

where $\varepsilon_{MG}$, $\varepsilon_{BAs}$, $f_{BC}$ and $\varepsilon_{BC}$ indicate mixture aerosol dielectric function, BAs complex dielectric function, BC volume fraction and BC complex dielectric function, respectively. For $\varepsilon_{BAs}$ and $\varepsilon_{BC}$ values can be obtained through RI:

$$\varepsilon_j=RI_j^2 \tag{3}$$

where $j$= BC, BAs, represents different component.

After obtaining $\varepsilon_{MG}$, the mixture aerosol RI can be calculated by Eqs. (4) and (5):

$$n=\sqrt{\left(\sqrt{\varepsilon_r^2+\varepsilon_i^2}+\varepsilon_r\right)\Big/2} \tag{4}$$

$$k=\sqrt{\left(\sqrt{\varepsilon_r^2+\varepsilon_i^2}-\varepsilon_r\right)\Big/2} \tag{5}$$

where $\varepsilon_r$ and $\varepsilon_i$ are real part and imaginary part of $\varepsilon_{MG}$, $n$ and $k$ are real part and imaginary part of the mixture aerosol RI.

We used the $RI_{BC}=1.95-0.79i$ (Bond and Bergstrom, 2006). As regards $RI_{BAs}$, we obtained them from AERONET based on K-means cluster method (Russell et al., 2014). In AERONET V3 daily product, we used the following criteria to remove strong absorbing fine aerosol particles data before clustering: (1) At a wavelength of 440 nm, many aerosol particles exhibit strong absorption, making it challenging to separate strongly absorbing BC particles. However, BC still exhibits strong




absorption in the 675-1020 nm range, and this characteristic can be leveraged to effectively isolate strongly absorbing fine particles (Bond et al., 2013), so we choose to remove the data with SSA (675-1020nm) < 0.85 and Fine mode fraction (FMF) > 0.4; (2) In some biomass combustion and industrial cases, SSA values range from 0.85 to 0.95 (Dubovik et al., 2002). In order to reduce the impact of BC aerosols on classification, fine particles whose SSA decreases with wavelength

climbing (Ångström Exponent, AE>1.5) are also removed. Fig. 3 shows the particle volume size distribution and SSA of BAs at different times. It can be observed that during spring and summer, the volume concentration of coarse-mode particles is higher in BAs, which is associated with the frequent occurrence of dust aerosols from North Africa during this period (Meloni et al., 2008). Moreover, the changes in SSA across different seasons are quite pronounced, with the absorption of fine aerosol particles being higher in winter. Using data from all seasons for clustering could introduce significant errors in

the estimation of BAs. Therefore, this study clustered the AERONET data by season to obtain accurate seasonal variations in the physical properties of BAs. The corresponding clustering results are presented in Tables 2 and 3.

**Table 2.** RI clustering results of BAs in different seasons.

| Time | 0.440μm | 0.675μm | 0.870μm | 1.020μm |
|------|---------|---------|---------|---------|
| DJF | 1.429-0.005$i$ | 1.434-0.004$i$ | 1.433-0.004$i$ | 1.428-0.004$i$ |
| MAM | 1.448-0.004$i$ | 1.453-0.003$i$ | 1.453-0.003$i$ | 1.448-0.003$i$ |
| JJA | 1.443-0.004$i$ | 1.455-0.003$i$ | 1.455-0.003$i$ | 1.452-0.003$i$ |
| SON | 1.430-0.004$i$ | 1.434-0.003$i$ | 1.435-0.004$i$ | 1.432-0.004$i$ |

**Table 3.** Particle volume size distribution parameters clustering results of BAs in different seasons. Vol-m, VMR-m, and

Std-m (m = F, C; F = Fine mode, C = Coarse mode) represent particle volume concentration, volume median radius, and standard deviation, respectively.

| Time | Vol-F | Vol-C | VMR-F | VMR-C | Std-F | Std-C | FMF |
|------|-------|-------|-------|-------|-------|-------|-----|
| DJF | 0.077 | 0.071 | 0.224 | 2.925 | 0.544 | 0.611 | 0.520 |
| MAM | 0.061 | 0.122 | 0.192 | 2.545 | 0.539 | 0.624 | 0.333 |
| JJA | 0.057 | 0.129 | 0.164 | 2.535 | 0.505 | 0.614 | 0.306 |
| SON | 0.070 | 0.081 | 0.214 | 2.894 | 0.508 | 0.603 | 0.464 |

In the 6SV2.1 model, we need to input aerosol mixture RI and particle volume size distribution. The particle volume size distribution equation is as follows (Dubovik and King, 2000):

$$dV/dlnr = \sum_{i=1}^{n}\left(C_i/\sqrt{2\pi}ln\sigma_i\right)exp\left[-0.5\left(\left(ln\,r - ln\,r_{m,i}\right)/ln\,\sigma_i\right)^2\right] \quad (6)$$

In Eq. (6), $i$ represents components, including BC, fine BAs, and Coarse BAs; $r$ represents particle radius; $C_i$, $\sigma_i$, and $r_{m,j}$ represent particles volume concentration, standard deviation, and volume median radius of different components,



respectively. Particles volume size distribution parameters for BAs have been shown in Table 3. For BC, $r_{m,BC} = 0.095\mu m$ and $\sigma_{BC}=1.80\mu m$ (Ganguly et al., 2009). Because the $C_{total}$ is normalization parameter, $C_i$ is equal to the volume fraction of each component.

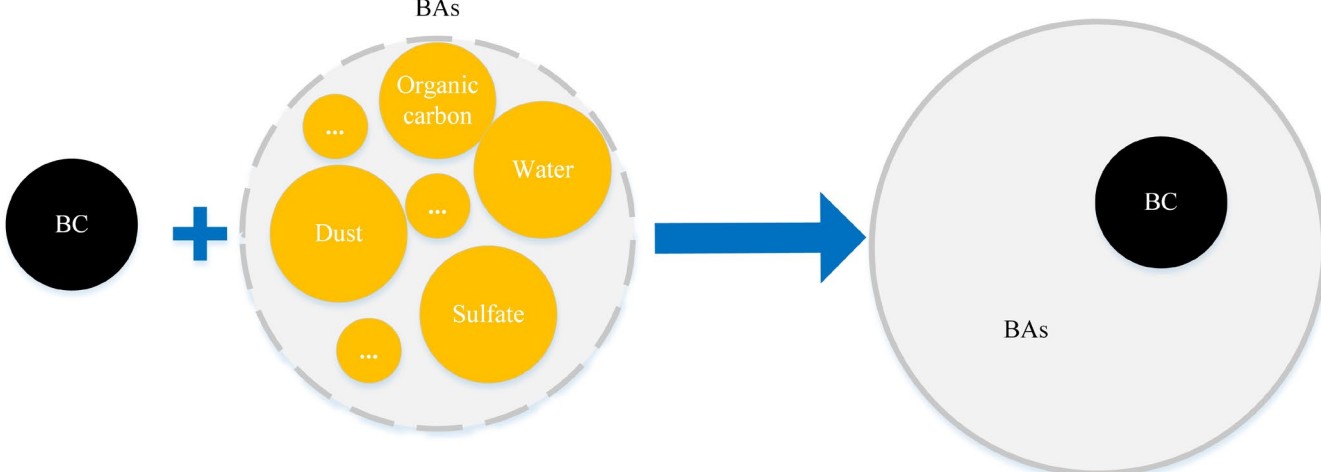

**Fig. 2.** Example diagram of internal mixing of aerosols.

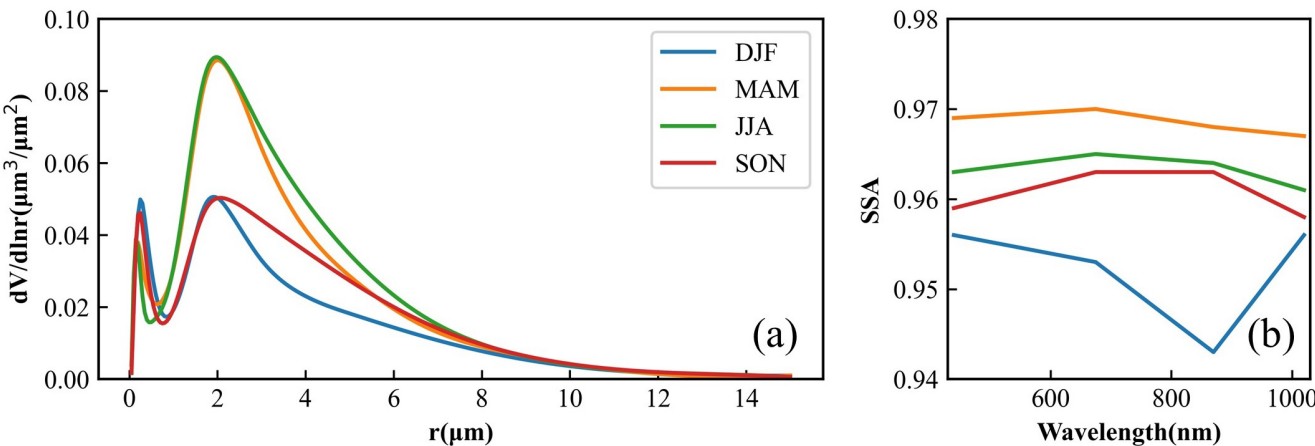

**Fig. 3.** BAs particle optical properties in different seasons. (a) and (b) denote particle volume size distribution and SSA, respectively. DJF represents winter (December-January-February); MAM represents spring (March-April-May); JJA represents summer (June-July-August); SON represents autumn (September- October-November).

### 3.3 Inverse method

After geometric correction of the MODIS L1B data, the water vapor and ozone data of MERRA-2 were used to correct the gas absorption of the band (Xie et al., 2020). Considering that at 0.47 µm, there is strong absorption by other particles (like BrC et al.) besides BC absorption (Chung et al., 2012), this study used the band of 0.66µm and 2.12µm for inversion. The





cloud mask algorithm used multiple wavelengths from visible to near-infrared for cloud identification (Xue et al., 2014). Then we used the 6SV2.1 to build LUT based on mixture aerosol optical properties. Since the proportion of BC in mixture aerosol particles generally does not exceed 6%, when generating the LUT, the value of $f_{BC}$ is from 0 to 0.06, and the step size is 0.01. In the retrieval process, we used DT AOD to input for finding optimal results, the cost function is as follows:

$$\chi = min \sum_{j=1}^{2} \left( \rho_{s,j}^{cal} - \rho_{s,j}^{DT} \right)^2 \tag{7}$$

where $j$ represents 2 band (0.66μm and 2.12μm), $\rho_s^{cal}$ and $\rho_s^{DT}$ are calculated surface reflectance and surface reflectance generated by DT algorithm, respectively.

Once the $f_{BC}$ is obtained, the BC column concentration can be calculated according to the following equation:

$$BC_{column} = f_{BC} * C_{total} * \rho_{BC} \tag{8}$$

$$C_{total} = C_{BAs}(\tau)/(1 - f_{BC}) \tag{9}$$

In Eq. (8), $\rho_{BC}$= 1.0 g/cm$^3$ (Ganguly et al., 2009), represents mass density of BC particles. $C_{BAs}$ is the integral of the volume size distribution of BAs obtained from AERONET clustering.

Since AE33 measures BC surface concentration ($BC_{surface}$), the inverted $BC_{column}$ needs to be converted. Previous studies assumed that BC was continuous uniform distribution below the atmospheric boundary layer, and directly divided $BC_{column}$ by the boundary layer height to obtain $BC_{surface}$ (Li et al., 2020; Bao et al., 2019). However, it is well known that the vertical distribution of BC is not uniform (Yuan et al., 2022), so this study used MERRA-2 data to obtain the ratio $K$ at each pixel to improve the accuracy of the conversion. The conversion equation is as follows:

$$BC_{surface} = K * BC_{column} \tag{10}$$

The overall inversion process is shown in Fig. 4.






**Fig. 4.** BC surface concentration retrieval algorithm workflow chart.

## 3.4 Model sensitivity analysis

Based on the aerosol model physical properties obtained above, we took the aerosol model in the DJF period as an example
to conduct sensitivity analysis on 6SV2.1. The solar zenith angle, satellite zenith angle and relative azimuth angle are 30°, 30°
and 12° respectively, and the surface reflectance of 0.66μm change steps are 0.02, 0.04, 0.06, 0.08, and 0.1. The relevant
results are shown in Fig. 5.

As shown in Figs. 5(a)-(b), the 6SV2.1 model is highly sensitive to variations in $f_{BC}$ in most cases. In dark surface areas ($\rho_s$=
0.02), the TOA signal changes as AOD increases, with the change becoming more pronounced as the $f_{BC}$ decreases. In bright
surface areas ($\rho_s$= 0.1), when $f_{BC} > 0.04$ and AOD > 1.5, TOA becomes insensitive to further increases in AOD, indicating
that in regions with high aerosol load and high $f_{BC}$, the inversion algorithm's performance significantly deteriorates. In Fig.
5(c), as the BC fraction increases, the SSA, which is independent of aerosol load, decreases notably, suggesting that BC





content has a substantial impact on the overall aerosol absorption properties. Additionally, the trend of the TOA standard deviation reveals that under identical surface conditions, high aerosol load results in a higher TOA standard deviation compared to low aerosol load. This implies that the model's sensitivity to BC inversion improves with higher AOD.

Furthermore, at high aerosol loading (AOD ≥ 1.0), the model estimates a higher TOA standard deviation at lower surface reflectance, indicating better performance in BC estimation under these conditions.



**Fig. 5.** Sensitivity analysis of BC inversion based on 6SV2.1. (a)-(b) represent $\rho_s$ at 0.66μm change steps are 0.02 and 0.10, respectively. (c) represents SSA and Standard deviation of TOA changes in different $f_{BC}$.

## 4. Results and discussion

### 4.1 Inverse results

Fig. 6 shows the monthly variations in MODIS BC surface concentration from November 2023 to June 2024. It is evident that in the regions surrounding northern Italy, the BC concentration exhibits a pattern of first increasing and then decreasing from November 2023 to March 2024, with emission levels significantly higher than in other areas. This trend is likely related to the region's high population density, developed industry, and low temperatures, which hinder the timely dispersion of emitted BC. From April to June, the overall BC concentration in the study area remains at a relatively low level.

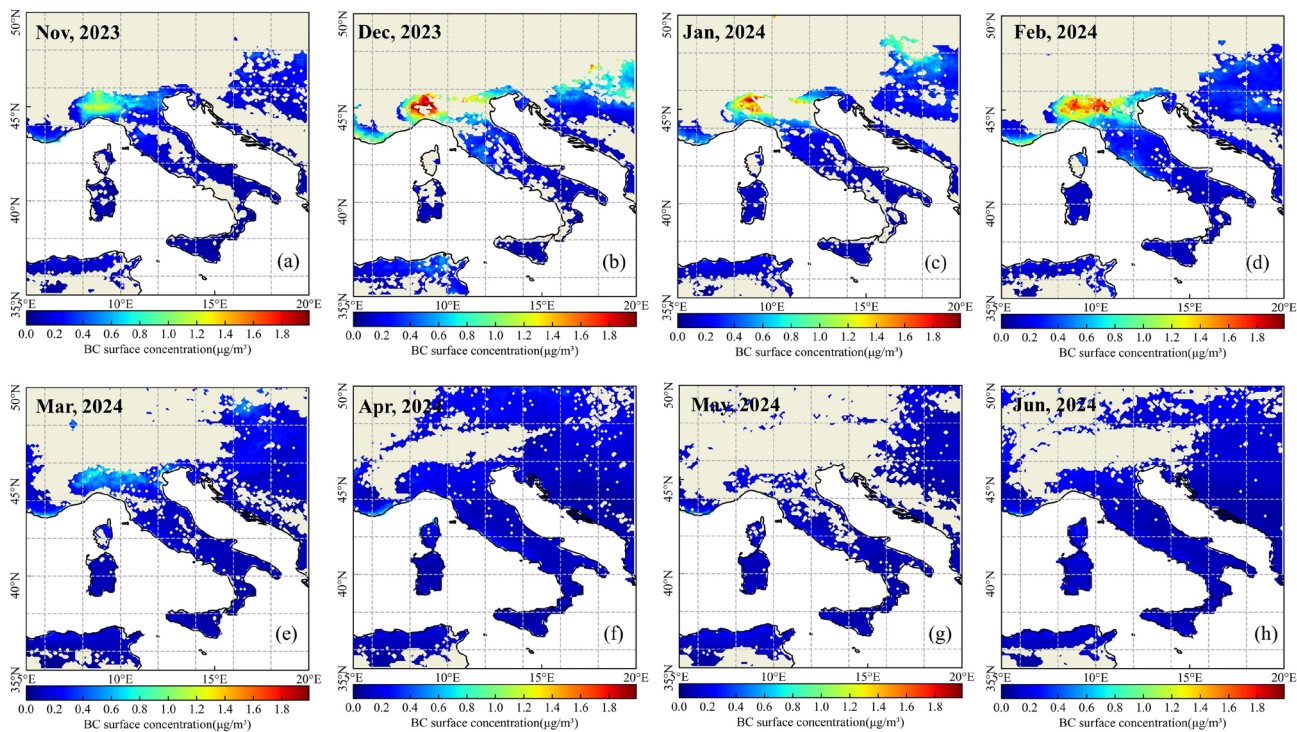

**Fig. 6.** Maps of monthly BC surface concentration distribution. (a)-(h) respectively represent from November 2023 to June 2024.





## 4.2 Validation


To specifically compare the differences between MODIS BC and AE33 BC, this study employed a spatio-temporal matching method. For MODIS, the average value of valid pixel data within a 50 km radius centered on the location of the ground station was used. For AE33, the average value was calculated from the data within one hour before and after the satellite's transit time. (Remer et al., 2005). Fig. 7 presents a comparison of the trend changes in the observed valid values of MODIS

BC and AE33 BC from November 2023 to June 2024 at three AE33 stations (ISAC Bologna II, Marseille Longchamp, and Milano Pascal) located in high BC emission areas. It is evident that the fluctuation trends of MODIS BC and AE33 BC are generally consistent, although MODIS BC tends to be lower than AE33 BC most of the time. Therefore, the inversion results based on this algorithm can accurately capture the spatiotemporal variations of BC in high-emission areas.

Fig. 8(a) presents a scatter plot of all valid values between MODIS BC and the six AE33 BC sites located in the study area

from November 2023 to June 2024, used to quantitatively evaluate the accuracy of the algorithm. Additionally, Fig. 8(b) shows the BC surface concentration verification accuracy of MERRA-2 for comparison. Statistical parameters include the total number of matching points (N), correlation coefficient (R), root mean square error (RMSE), mean absolute error (MAE), and linear fitting function (Jiang et al., 2024). The corresponding accuracy validation results are summarized in Table 4. The correlation coefficient (R) for MODIS BC is 0.727, while for MERRA-2 BC it is 0.655, indicating that our

algorithm performs better in terms of relevance. The RMSE for MODIS BC is 0.353, compared to 0.487 for MERRA-2 BC, and the MAE for MODIS BC is 0.211, whereas for MERRA-2 BC it is 0.381. These results suggest that the numerical difference between MODIS BC and AE33 is smaller, indicating better accuracy for MODIS BC. From the perspective of linear regression and scatter point density, MERRA-2 tends to overestimate the surface concentration of BC, while MODIS BC shows the opposite trend. This may be related to the MG-EMA model only considers BC internal mixing state, but there

may still be a small amount of fresh and exposed BC externally mixed in the atmosphere (China et al., 2013), which may result in an underestimate of BC.



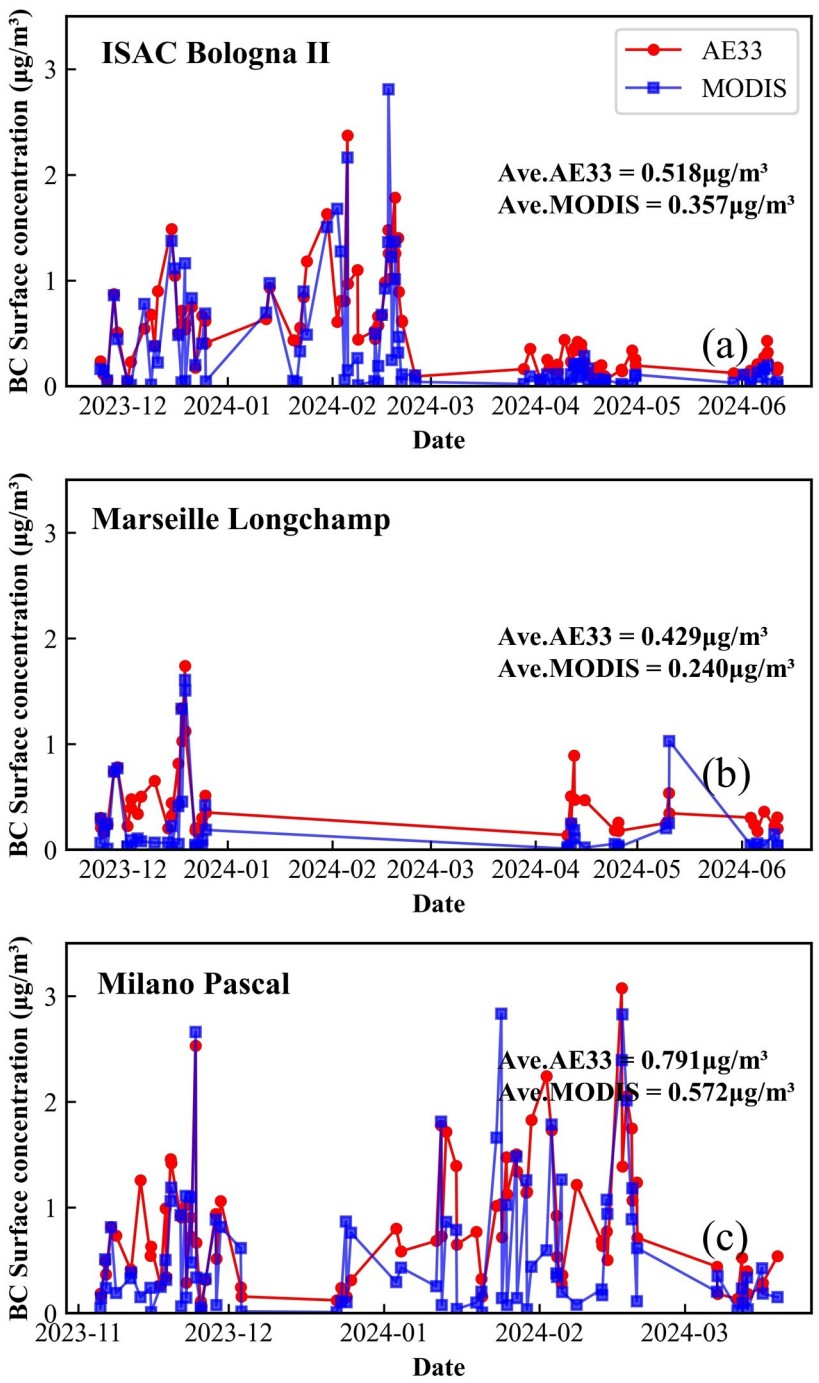

**Fig. 7.** Comparison of consistency changes between MODIS BC and AE33 BC. (a), (b), and (c) represent ISAC Bologna II, Marseille Longchamp, and Milano Pascal, respectively.



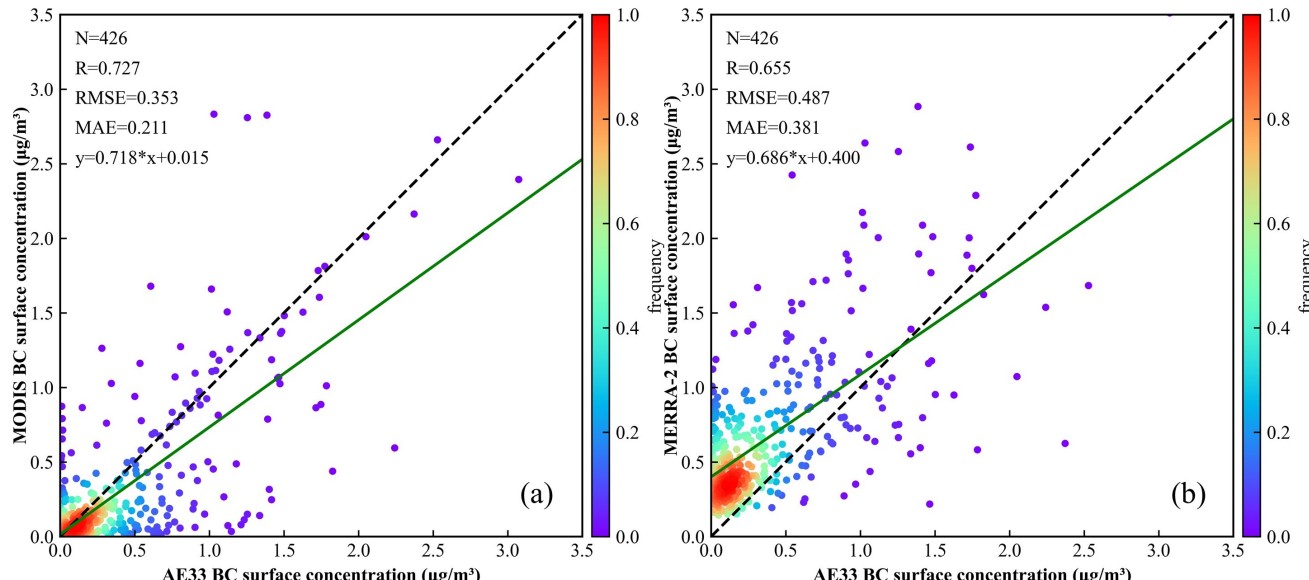


**Fig. 8.** Scatter plot of MODIS BC and AE33 versus MERRA-2 BC and AE33. (a) and (b) represent MODIS BC and MERRA-2 BC, respectively. The black dashed line and the green solid line represent the 1:1 line and the linear regression function respectively.

**Table 4.** Summary of statistical parameter results from Fig. 8.

| Statistical parameter | MODIS BC | MERRA-2 BC |
|:---:|:---:|:---:|
| N | 426 | 426 |
| R | 0.727 | 0.655 |
| RMSE | 0.353 | 0.487 |
| MAE | 0.211 | 0.381 |
| Slope | 0.718 | 0.686 |
| Offset | 0.015 | 0.400 |

**4.3 Uncertainty analysis**

Fig. 9 analyzes the influence of AOD and surface reflectance on the bias (MODIS - AE33) of the retrieved BC surface concentration. The results reveal that the BC bias exhibits different patterns of change. As shown in Fig. 9(a), when AOD is low, the uncertainty of the bias is high, but as AOD increases, the bias decreases. When AOD > 0.75, the overall bias approaches zero, and the uncertainty becomes very low. This trend aligns with the conclusion in Section 3.4, where the

retrieval accuracy improves under high AOD conditions due to the stronger aerosol signal. In Fig. 9(b), the uncertainty of the bias gradually increases as surface reflectance rises. When $\rho_s > 0.08$, the uncertainty of the bias increases significantly,





suggesting that the algorithm's applicability in bright surface areas still needs improvement. However, it is worth noting that due to the lack of data from AE33 stations in high brightness areas, the surface reflectance of the AE33 stations in this study is below 0.12, and DT AOD accuracy is better in dark surface, so further research is needed to determine the applicability of

bright surface reflectance. Therefore, the uncertainty analysis confirms that the algorithm performs better in conditions of high AOD.

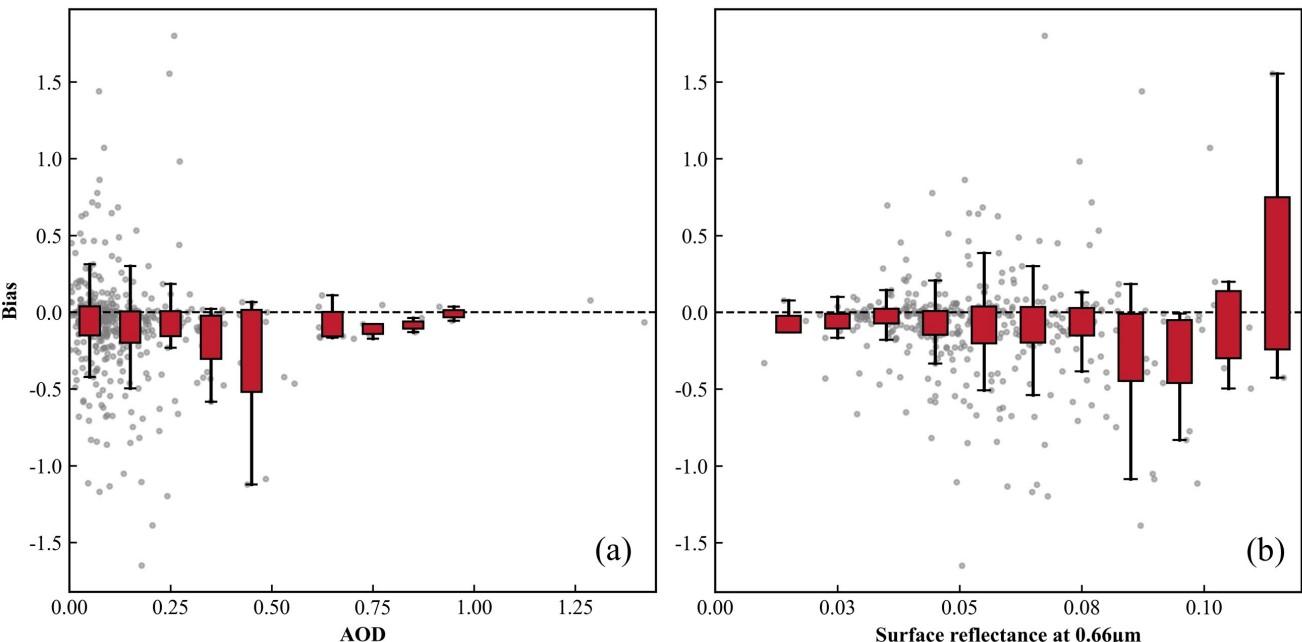

**Fig. 9.** BC surface concentration bias (MODIS - AE33) dependence analysis on the (a) AOD and (b) surface reflectance.

## 5. Conclusion

In this paper, we proposed a new algorithm for retrieving BC surface concentrations based on joint MODIS and AERONET data. First, the algorithm derived the optical properties of internally mixed BAs for each season from the AERONET V3 daily product. It used the particle volume size distribution and MG-MEA approximation equation to calculate the RI and volume concentration of the internal mixed aerosol, which were the 6SV2.1 input data of the aerosol model. Then, the sensitivity analysis was conducted for AOD, surface reflectance, and BC fraction. During the inversion process, multiple

LUTs were established based on different BC fration, and DT AOD values were inputted to iteratively find the optimal $f_{BC}$. Next, the BC column concentration and surface concentration values generated by MERRA-2 were used to convert the inverted BC column concentration to surface concentration. Finally, the retrieved BC surface concentrations were validated against AE33 observation data. The BC surface concentrations obtained by this algorithm show relatively high reliability and accuracy (R = 0.727, RMSE = 0.353, MAE = 0.211), though there is a slight overall underestimation (y = 0.718x + 0.015)



compared to high-precision ground-based in-situ measurements. Additionally, uncertainty analysis of the inversion results indicates that the algorithm is more suitable for high AOD and dark surface conditions. Therefore, future work will focus on improving the algorithm's performance in bright surface and low aerosol loading areas.

**Author contribution**

Xingxing Jiang: Data curation, Methodology, Software, Writing– original draft & editing. Yong Xue: Conceptualization,
Writing - review & editing. Mariarosaria Calvello: Review & editing. Shuhui Wu: Data download. Pei Li: Data download. All authors have read and agreed to the published version of the manuscript.

**Competing interests**

The authors declare that they have no conflict of interest.

**Acknowledgements**

The authors gratefully acknowledge the NASA teams for providing MODIS, AERONET, and MERRA-2 data, and the EBAS group in the Norwegian Institute for Air Research for delivering an open-access AE33 BC surface concentration database.

**Financial support**

This work was partly supported by the National Natural Science Foundation of China (NSFC) under Grant No. 42275147
and the China Scholarship Council (CSC) Grant No. (202306420061).

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
