# Peer review of "Retrieval of Black Carbon Aerosol Surface Concentration Using Integrated MODIS and AERONET Data"

_EGUsphere, 2025_

## Author Comment (AC1)

Dear Editors and Reviewers,

Thanks very much for taking your time to review this manuscript. We really appreciate all your detailed valuable comments on our manuscript of "egusphere-2025-435". These suggestions have been so helpful that we have incorporated them into the newly revised manuscript. And we hope that the reviewers and the editors will be satisfied with our responses to the 'comments' and the revisions for this manuscript. Please find our itemized responses in below and my revisions in the re-submitted files.

Yours Sincerely,

Yong Xue

**Response for Reviewer #1:**

This manuscript presents a novel algorithm for retrieving black carbon (BC) surface concentrations by synergizing MODIS and AERONET data, integrating K-means clustering, Maxwell-Garnett effective medium approximation (MG-EMA), 6SV2.1 radiative transfer modeling, and MERRA-2-based vertical conversion. Seasonal aerosol modeling via K-means clustering and the MG-EMA significantly improves the representation of internally mixed aerosols. Validation against AE33 in-situ measurements (R = 0.727, RMSE = 0.353) demonstrates strong agreement, while comparison with MERRA-2 BC data effectively highlights the algorithm's superior performance. Besides, the uncertainty analysis confirms that the algorithm performs better in conditions of high AOD.

Overall, the manuscript is well-structured and scientifically sound. Due to MODIS having continuous long-term observational data (1999 to present), the results of this study contribute to obtaining a more accurate and long-term series of BC surface concentration datasets, providing richer reference information for climate change and air quality research. Therefore, I suggest accepting the manuscript for publication with slight modifications. Please find my specific suggestions below.

Overall, the method used in this manuscript is unique and innovative. The above are my comments and opinions. I look forward to the author's reply.

**Response:** Thank you very much for these positive comments and please find our detailed responses below to all your suggestions.

Specific Comments:
1. In section 1, the introduction on satellite platform inversion of BC only lasts until 2023. It is recommended to supplement the latest research progress in this field.
**Response:** Thank you for your suggestion. We have added new research literature the Introduction part of the manuscript.

2. In Fig.3(a), it is difficult to distinguish the particle size distribution of each season in the fine mode region using equidistant radius r scales. It is recommended to use logarithmic scales to increase discrimination.
**Response:** Thank you for your suggestion. We have changed the radius (r) coordinate scale of Fig.3 (a) to log.

[Figure]

3. In Fig.4, the spelling of 'reteieval' is incorrect and needs to be changed to 'retrieval'.

**Response:** Thank you very much for pointing out this issue. We have already replaced 'reteieval' in Fig. 4 with 'retrieval'.

4. In section 3.4, there are aerosol optical property data for DJF, MAM, JJA, and SON. Why choose DJF for sensitivity analysis?

**Response:** The reasons for choosing the DJF period's aerosol model for sensitivity analysis in the manuscript are as follows: (1) The black carbon concentration is relatively high in winter, making the data of this period more representative for the sensitivity analysis of the inversion algorithm. (2) The BAs constructed based on AERONET V3 daily data show significant differences in different seasons, and the absorption characteristics of DJF are more obvious, which is conducive to revealing the performance of the model, especially when the aerosol concentration is high.

5. In section 3.4, only the sensitivity of surface reflectance less than 0.1 is analyzed, and most bright surfaces (such as deserts) are not included. It is recommended to expand the numerical range and set the threshold below 0.3.

**Response:** Thank you for your suggestion. We have adjusted the contents in Fig. 5, analyzed the performance at surface reflectance of 0.20 and 0.30, and revised the expressions in the text. "As shown in Fig. 5(a)-(d), the sensitivity analysis results indicate that as AOD increases, the estimated TOA standard deviation under different surface conditions gradually increases, suggesting that the theoretical inversion accuracy is higher under high aerosol loading conditions. However, when $\rho_s$ = 0.10 and the aerosol loading is high (AOD > 1.0), if the BC fraction is high ($f_{BC} \geq 0.04$), the TOA will basically not change with the increase of AOD, which will lead to an unsatisfactory inversion effect under such conditions. In Fig. 5(e), as the BC fraction increases, the SSA, which is independent of aerosol load, decreases notably, suggesting that BC content has a substantial impact on the overall aerosol absorption properties. Additionally, under low $f_{BC}$ conditions, the standard deviation of the estimated TOA for dark surfaces is higher, while under high $f_{BC}$ conditions, the standard deviation of the estimated TOA for bright surfaces is higher. This indicates that bright surfaces are more sensitive to absorbing aerosols and are more conducive to estimating strongly absorbing BC particles."

6. In section 4.1, "This trend is likely related to the region's high population density, developed industry, and low temperatures, which hinder the timely dispersion of emitted BC. From April to June, the overall BC concentration in the study area remains at a relatively low level". I suggest adding references here to prove that these factors do indeed affect the high values.

**Response:** Additionally, the presence of the northern and western Alps as well as the southern Apennine Mountains determines weak wind conditions and frequent temperature retrogrades, which hinder atmospheric diffusion and trap pollution on the ground (Renna et al., 2024).

Renna, S., J. Lunghi, F. Granella, M. Malpede, and D. Di Simine, 2024: Impacts of agriculture on PM$_{10}$ pollution and human health in the Lombardy region in Italy. *Front. Environ. Sci.*, **12**, 1369678, doi: 10.3389/fenvs.2024.1369678.

7. In section 4.2, "It is evident that the fluctuation trends of MODIS BC and AE33 BC are generally consistent, although MODIS BC tends to be lower than AE33 BC most of the time." Why is there a trend of underestimation?

**Response:** This may be related to the MG-EMA model which only considers BC internal mixing state, but there may still be a small amount of fresh and exposed BC externally mixed in the atmosphere (China et al., 2013), which may result in an underestimate of BC.

China, S., C. Mazzoleni, K. Gorkowski, A. C. Aiken, and M. K. Dubey, 2013: Morphology and mixing state of individual freshly emitted wildfire carbonaceous particles. *Nat. Commun.*, **4**(1), 2122, doi: 10.1038/ncomms3122.

8. In Fig.7, the author only presented data from 3 out of 6 AE33 sites, and it may be considered to display data from other sites.

**Response:** The satellite-ground comparison data of the three AE33 sites (Zurich-Kaserne, Monte Cimone, and Kosetice (NAOK)) are shown in the following figure. The number of valid matching values of Zurich-Kaserne and Kosetice (NAOK) is very small. Although the number of valid matching values of Monte Cimone is large, the BC concentration has remained at a relatively low level, and the degree of change is not as obvious as that of the three sites (ISAC Bologna II, Marseille Longchamp and Milano Pascal) shown in Fig. 7. Therefore, it is not shown in the manuscript.

[Figure]

9. In section 4.3, "suggesting that the algorithm's applicability in bright surface areas still needs improvement. However, it is worth noting that due to the lack of data from AE33 stations in high brightness areas, the surface reflectance of the AE33 stations in this study is below 0.12, and DT AOD accuracy is better in dark surface, so further research is needed to determine the applicability of bright surface reflectance." The surface reflectance here is relatively low, making it difficult to reveal the characteristics of bright surfaces. It is recommended to revise the statement to make it more rigorous.

**Response:** We have made the following modifications: "When $\rho_s$ >0.08, the uncertainty of the bias increases significantly, suggesting that the algorithm's applicability in relatively high surface reflectance areas still needs improvement. However, it is worth noting that due to the lack of data from AE33 stations in high brightness areas ($\rho_s \geq 0.2$), the surface reflectance of the AE33 stations in this study is below 0.12, and when $\rho_s$ > 0.1, the data also is insufficient. Therefore, uncertainty analysis confirms that this retrieval algorithm has better performance under high AOD conditions. However, due to the lack of ground-based AE33 observation data in high-brightness surface areas, the accuracy under this surface condition still lacks effective validation."

---

## Author Comment (AC2)

Dear Editors and Reviewers,

Thanks very much for taking your time to review this manuscript. We really appreciate all your detailed valuable comments on our manuscript of "egusphere-2025-435". These suggestions have been so helpful that we have incorporated them into the newly revised manuscript. And we hope that the reviewers and the editors will be satisfied with our responses to the 'comments' and the revisions for this manuscript. Please find our itemized responses in below and my revisions in the re-submitted files.

Yours Sincerely,

Yong Xue

**Response for Reviewer #2:**

The authors developed a novel algorithm to retrieve BC surface concentration using satellite data through several steps, including determined the background aerosol model, the Maxwell–Garnett effective medium approximation model, lookup tables, and a conversion coefficient. The retrieved BC surface concentrations were validated by using ground measurements, and the statistical parameters outperform those obtained from MERRA-2 BC data. In general, this manuscript is well organized and written. Some explanations are needed for a clear understanding. Generally, this manuscript can be accepted for publication after minor revisions.

As for the proposed algorithms and models, it is suggested to introduce the physical and chemical mechanisms, and how to consider and apply them in model designs and improvements. What are the advantages and disadvantages of the algorithms and models in this study? They should be discussed somewhere in the text.

**Response:** Thank you very much for your positive comments. We have further described the advantages and disadvantages of this algorithm in the conclusion section: "The BC surface concentrations obtained by this algorithm show relatively high reliability and accuracy (R = 0.727, RMSE = 0.353, MAE = 0.211), though there is a slight overall underestimation (y = 0.718x + 0.015) compared to high-precision ground-based in-situ measurements. This might be due to a small number of BC particles being exposed on the outside of the shell, which led to the failure to estimate the relevant aspect. Additionally, uncertainty analysis of the inversion results indicates that the algorithm is more suitable for high AOD conditions. However, since there is no AE33 site data in the bright surface area, the performance of the inversion results on the bright surface still needs further verification. Therefore, future work will focus on improving the algorithm's performance low aerosol loading conditions and evaluating inversion results accuracy in bright surface." The specific responses to the comments are as follows:

1. L11-16, "In this study, we developed a novel algorithm for retrieving BC surface concentration using MODIS and AERONET data. The algorithm first determined the seasonal background aerosol model using the K-means clustering method, based on AERONET V3 daily products. It then employed the Maxwell–Garnett effective medium approximation model to calculate the complex refractive index of the internally mixed aerosols and used the 6SV2.1 radiative transfer code to establish lookup tables for optimal BC fraction and column concentration estimation", this paragraph should be rewritten.

**Response:** Thank you for your suggestion. We have rewritten this paragraph. "In this study, we developed a novel algorithm for retrieving BC surface concentration jointly using MODIS and AERONET data. Firstly, the algorithm employed the K-means clustering method to determine seasonal background aerosols model based on AERONET V3 daily products. Then, the Maxwell–Garnett effective medium approximation model was utilized to calculate the complex refractive index of the internally mixed aerosols. Subsequently, the lookup tables were established using the

6SV2.1 radiative transfer code to estimate optimal BC fraction and column concentration. Next, the column concentration data were converted to surface concentration using a conversion coefficient derived from MERRA-2. Finally, the retrieved MODIS BC surface concentration was validated with in-situ Aethalometer measurements."

2. L77, These datasets can be obtained from this website (https://ladsweb.modaps.eosdis.nasa.gov/), the accessed date should be added. It is same for all websites used in the text.
**Response:** Thank you very much for your suggestion. We have supplemented the data access dates in the manuscript.

3. L137-138, which is associated with the frequent occurrence of dust aerosols from North Africa during this period (Meloni et al., 2008), please explain the time period (e.g., year) and the sites, as well as in Tables 2 and 3.
**Response:** Thank you for your suggestion. We have made the following revisions to the manuscript: "It can be observed that during spring and summer, the volume concentration of coarse-mode particles is higher in BAs, which is associated with the frequent occurrence of dust aerosols from North Africa during from March to June every year (Meloni et al., 2008). Moreover, the changes in SSA across different seasons are quite pronounced, with the absorption of fine aerosol particles being higher in winter. Using data from all seasons for clustering could introduce significant errors in the estimation of BAs. Therefore, this study clustered the AERONET data by season to obtain accurate seasonal variations in the physical properties of BAs. Table 2 and Table 3 show RI clustering results and particle volume size distribution parameters clustering results of BAs in different seasons."

4. L167-168, since the proportion of BC in mixture aerosol particles generally does not exceed 6%, please explain where it is come from for 6%.
**Response:** Thank you for your suggestion. We have already added the data sources to the manuscript. "Since the fraction of BC in mixture aerosol particles generally does not exceed 6% (Bao et al., 2020)."

Bao, F., Li, Y., Cheng, T., Gao, J., and Yuan, S.: Estimating the Columnar Concentrations of Black Carbon Aerosols in China Using MODIS Products, Environ. Sci. Technol., 54, 11025-11036, https://doi.org/10.1021/acs.est.0c00816, 2020.

5. L200-201, Furthermore, at high aerosol loading (AOD ≥ 1.0), the model estimates a higher TOA standard deviation at lower surface reflectance, indicating better performance in BC estimation under these conditions. It is confused for that the model estimates a higher TOA standard deviation and better performance, more explanations are suggested.
**Response:** Under all the same conditions, if the estimated TOA differences under different FBCS are very small, that is, the standard deviation of TOA is very small, then

the cost function will have difficulty distinguishing such differences, thereby resulting in a large error in the estimation of $f_{BC}$. In Fig. 5, in the case of high AOD, the differences in TOA estimated by different $f_{BC}$ are obvious. At this time, the cost function will have a significant difference, thereby improving the estimation accuracy of $f_{BC}$.

6. L203-204, Fig. 5. Sensitivity analysis of BC inversion based on 6SV2.1. (a)-(b) represent $\rho_s$ at 0.66μm change steps are 0.02 and 0.10, respectively, should be rewritten.

**Response:** We have rewritten it as follows: "**Fig. 5.** Sensitivity analysis of BC inversion based on 6SV2.1 model. The (a)-(d) represents the $\rho_s$ at 0.66μm variation step sizes, which are 0.02, 0.10, 0.20, and 0.30, respectively. The (e) represents SSA and Standard deviation of TOA changes in different $f_{BC}$."

7. Fig. 9. BC surface concentration bias (MODIS - AE33) dependence analysis on the (a) AOD and (b) surface reflectance, please indicate the meanings of the bars.

**Response:** Thank you for your suggestion. We have already modified the title of Fig. 9: **Fig.9.** The box plot of BC surface concentration Bias (MODIS - AE33) independence analysis on the (a) AOD and (b) surface reflectance. The red box represents the interquartile range (IQR, 25th–75th percentiles), and the black whiskers extend to the most extreme data points within 1.5×IQR from the quartiles.

8. L224-236, it is recommended to make more comparisons with other studies, including the main statistical parameters shown in the text.

**Response:** Since there is no inversion results of other algorithms based on MODIS data at the same time and place, we don't verify the inversion products with other algorithms. However, in the manuscript, we compare the accuracy of the inversion results with that of MERRA-2 BC. The comparison results indicated that the inversion accuracy of this algorithm is better.

9. L219, transit time. (Remer et al., 2005), please remove the dot after time.

**Response:** Thank you for pointing out this issue. We have removed the redundant dot from the manuscript.

---

## Author Response (AR1)

Dear Editors and Reviewers,

Thanks very much for your satisfaction with our responses to the 'comments' and the revisions for this manuscript.

Yours Sincerely,

Yong Xue

---

## Author Response (AR2)

Dear Editors and Reviewers,

Thanks very much for your satisfaction with our responses to the 'comments' and the revisions for this manuscript. In the latest uploaded manuscript, we have changed the format of the figure title number from "Fig. *" to "Figure *". In addition, we have also revised the Short summary text, changing BC to the full name Black Carbon, and the text does not exceed 500 characters.

Yours Sincerely,

Yong Xue